# BabyAI: A Platform to Study the Sample Efficiency of Grounded Language Learning

**Maxime Chevalier-Boisvert**[*]
Mila, Université de Montréal

**Dzmitry Bahdanau**[*]
Mila, Université de Montréal
AdeptMind Scholar
Element AI

**Salem Lahlou**
Mila, Université de Montréal

**Lucas Willems**[†]
École Normale Supérieure, Paris

**Chitwan Saharia**[†]
IIT Bombay

**Thien Huu Nguyen**[‡]
University of Oregon

**Yoshua Bengio**
Mila, Université de Montréal
CIFAR Senior Fellow

## Abstract

Allowing humans to interactively train artificial agents to understand language instructions is desirable for both practical and scientific reasons. Though, given the lack of sample efficiency in current learning methods, reaching this goal may require substantial research efforts. We introduce the BabyAI research platform, with the goal of supporting investigations towards including humans in the loop for grounded language learning. The BabyAI platform comprises an extensible suite of 19 levels of increasing difficulty. Each level gradually leads the agent towards acquiring a combinatorially rich synthetic language, which is a proper subset of English. The platform also provides a hand-crafted bot agent, which simulates a human teacher. We report estimated amount of supervision required for training neural reinforcement and behavioral-cloning agents on some BabyAI levels. We put forward strong evidence that current deep learning methods are not yet sufficiently sample-efficient in the context of learning a language with compositional properties.

## 1 Introduction

How can a human train an intelligent agent to understand natural language instructions? We believe that this research question is important from both technological and scientific perspectives. No matter how advanced AI technology becomes, human users will likely want to customize their intelligent helpers to better understand their desires and needs. On the other hand, developmental psychology, cognitive science and linguistics study similar questions but applied to human children, and a synergy is possible between research in grounded language learning by computers and research in human language acquisition.

In this work, we present the BabyAI research platform, whose purpose is to facilitate research on grounded language learning. In our platform we substitute a simulated human expert for a real human; yet our aspiration is that BabyAI-based studies enable substantial progress towards putting an actual human in the loop. The current domain of BabyAI is a 2D gridworld in which synthetic natural-looking instructions (e.g. "put the red ball next to the box on your left") require the agent to navigate the world (including unlocking doors) and move objects to specified locations. BabyAI improves upon similar prior setups (Hermann et al., 2017; Chaplot et al., 2018; Yu et al., 2018) by supporting simulation of certain essential aspects of the future human in the loop agent training:

---

[*]Equal contribution.
[†]Work done during an internship at Mila.
[‡]Work done during a post-doc at Mila.

*curriculum learning* and *interactive teaching*. The usefulness of curriculum learning for training machine learning models has been demonstrated numerous times in the literature (Bengio et al., 2009; Kumar et al., 2010; Zaremba and Sutskever, 2015; Graves et al., 2016), and we believe that gradually increasing the difficulty of the task will likely be essential for achieving efficient human-machine teaching, as in the case of human-human teaching. To facilitate curriculum learning studies, BabyAI currently features 19 levels in which the difficulty of the environment configuration and the complexity of the instruction language are gradually increased. Interactive teaching, i.e. teaching differently based on what the learner can currently achieve, is another key capability of human teachers. Many advanced agent training methods, including DAGGER (Ross et al., 2011), TAMER (Warnell et al., 2017) and learning from human preferences (Wilson et al., 2012; Christiano et al., 2017), assume that interaction between the learner and the teacher is possible. To support interactive experiments, BabyAI provides a bot agent that can be used to generate new demonstrations on the fly and advise the learner on how to continue acting.

Arguably, the main obstacle to language learning with a human in the loop is the amount of data (and thus human-machine interactions) that would be required. Deep learning methods that are used in the context of imitation learning or reinforcement learning paradigms have been shown to be very effective in both simulated language learning settings (Mei et al., 2016; Hermann et al., 2017) and applications (Sutskever et al., 2014; Bahdanau et al., 2015; Wu et al., 2016). These methods, however, require enormous amounts of data, either in terms of millions of reward function queries or hundreds of thousands of demonstrations. To show how our BabyAI platform can be used for sample efficiency research, we perform several case studies. In particular, we estimate the number of demonstrations/episodes required to solve several levels with imitation and reinforcement learning baselines. As a first step towards improving sample efficiency, we additionally investigate to which extent pretraining and interactive imitation learning can improve sample efficiency.

The concrete contributions of this paper are two-fold. First, we contribute the BabyAI research platform for learning to perform language instructions with a simulated human in the loop. The platform already contains 19 levels and can easily be extended. Second, we establish baseline results for all levels and report sample efficiency results for a number of learning approaches. The platform and pretrained models are available online. We hope that BabyAI will spur further research towards improving sample efficiency of grounded language learning, ultimately allowing human-in-the-loop training.

## 2 RELATED WORK

There are numerous 2D and 3D environments for studying synthetic language acquistion. (Hermann et al., 2017; Chaplot et al., 2018; Yu et al., 2018; Wu et al., 2018). Inspired by these efforts, BabyAI augments them by uniquely combining three desirable features. First, BabyAI supports world state manipulation, missing in the visually appealing 3D environments of Hermann et al. (2017), Chaplot et al. (2018) and Wu et al. (2018). In these environments, an agent can navigate around, but cannot alter its state by, for instance, moving objects. Secondly, BabyAI introduces partial observability (unlike the gridworld of Bahdanau et al. (2018)). Thirdly, BabyAI provides a systematic definition of the synthetic language. As opposed to using instruction templates, the Baby Language we introduce defines the semantics of all utterances generated by a context-free grammar (Section 3.2). This makes our language richer and more complete than prior work. Most importantly, BabyAI provides a simulated human expert, which can be used to investigate human-in-the-loop training, the aspiration of this paper.

Currently, most general-purpose simulation frameworks do not feature language, such as PycoLab (DeepMind, 2017), MazeBase (Sukhbaatar et al., 2015), Gazebo (Koenig and Howard, 2004), Viz-Doom (Kempka et al., 2016), DM-30 (Espeholt et al., 2018), and AI2-Thor (Kolve et al., 2017). Using a more realistic simulated environment such as a 3D rather than 2D world comes at a high computational cost. Therefore, BabyAI uses a gridworld rather than 3D environments. As we found that available gridworld platforms were insufficient for defining a compositional language, we built a MiniGrid environment for BabyAI.

General-purpose RL testbeds such as the Arcade Learning Environment (Bellemare et al., 2013), DM-30 (Espeholt et al., 2018), and MazeBase (Sukhbaatar et al., 2015) do not assume a human-in-the-loop setting. In order to simulate this, we have to assume that all rewards (except intrinsic

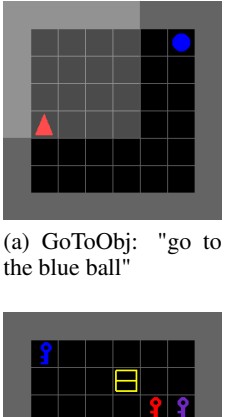

(a) GoToObj: "go to the blue ball"

(b) PutNextLocal: "put the blue key next to the green ball"

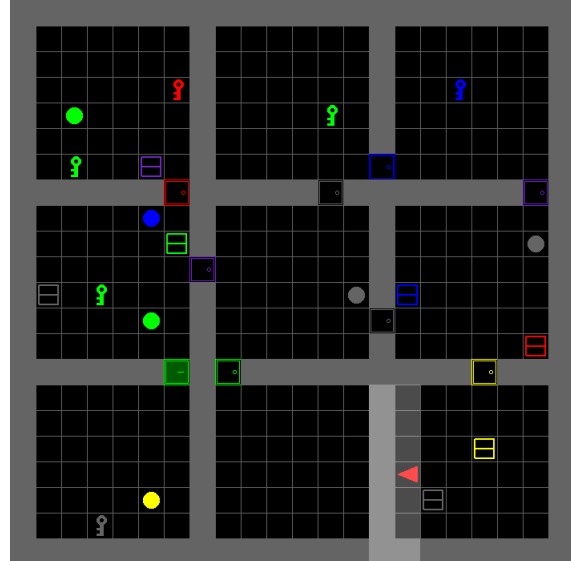

(c) BossLevel: "pick up the grey box behind you, then go to the grey key and open a door". Note that the green door near the bottom left needs to be unlocked with a green key, but this is not explicitly stated in the instruction.

Figure 1: Three BabyAI levels built using the MiniGrid environment. The red triangle represents the agent, and the light-grey shaded area represents its field of view (partial observation).

rewards) would have to be given by a human, and are therefore rather expensive to get. Under this assumption, imitation learning methods such as behavioral cloning, Searn (Daumé Iii et al., 2009), DAGGER (Ross et al., 2011) or maximum-entropy RL (Ziebart et al., 2008) are more appealing, as more learning can be achieved per human-input unit.

Similar to BabyAI, studying sample efficiency of deep learning methods was a goal of the bAbI tasks (Weston et al., 2016), which tested reasoning capabilities of a learning agent. Our work differs in both of the object of the study (grounded language with a simulated human in the loop) and in the method: instead of generating a fixed-size dataset and measuring the performance, we measure how much data a general-purpose model would require to get close-to-perfect performance.

There has been much research on instruction following with natural language (Tellex et al., 2011; Chen and Mooney, 2011; Artzi and Zettlemoyer, 2013; Mei et al., 2016; Williams et al., 2018) as well as several datasets including SAIL (Macmahon et al., 2006; Chen and Mooney, 2011) and Room-to-Room (Anderson et al., 2018). Instead of using natural language, BabyAI utilises a synthetic Baby language, in order to fully control the semantics of an instruction and easily generate as much data as needed.

Finally, Wang et al. (2016) presented a system that interactively learned language from a human. We note that their system relied on substantial amounts of prior knowledge about the task, most importantly a task-specific executable formal language.

## 3 BABYAI PLATFORM

The BabyAI platform that we present in this work comprises an efficiently simulated gridworld environment (MiniGrid) and a number of instruction-following tasks that we call *levels*, all formulated using subsets of a synthetic language (Baby Language). The platform also includes a bot that can generate successful demonstrations for all BabyAI levels. All the code is available online at https://github.com/mila-iqia/babyai/tree/iclr19.

### 3.1 MINIGRID ENVIRONMENT

Studies of sample efficiency are very computationally expensive given that multiple runs are required for different amounts of data. Hence, in our design of the environment, we have aimed for a minimalistic and efficient environment which still poses a considerable challenge for current general-purpose agent learning methods. We have implemented MiniGrid, a partially observable 2D gridworld environment. The environment is populated with entities of different colors, such as the agent, balls, boxes, doors and keys (see Figure 1). Objects can be picked up, dropped and moved around by the agent. Doors can be unlocked with keys matching their color. At each step, the agent receives a 7x7 representation of its field of view (the grid cells in front of it) as well as a Baby Language instruction (textual string).

The MiniGrid environment is fast and lightweight. Throughput of over 3000 frames per second is possible on a modern multi-core laptop, which makes experimentation quicker and more accessible. The environment is open source, available online, and supports integration with OpenAI Gym. For more details, see Appendix B.

### 3.2 BABY LANGUAGE

We have developed a synthetic Baby Language to give instructions to the agent as well as to automatically verify their execution. Although Baby Language utterances are a comparatively small subset of English, they are combinatorially rich and unambigously understood by humans. The language is intentionally kept simple, but still exhibits interesting combinatorial properties, and contains $2.48 \times 10^{19}$ possible instructions. In this language, the agent can be instructed to go to objects, pick up objects, open doors, and put objects next to other objects. The language also expresses the conjunction of several such tasks, for example "put a red ball next to the green box after you open the door". The Backus-Naur Form (BNF) grammar for the language is presented in Figure 2 and some example instructions drawn from this language are shown in Figure 3. In order to keep the resulting instructions readable by humans, we have imposed some structural restrictions on this language: the *and* connector can only appear inside the *then* and *after* forms, and instructions can contain no more than one *then* or *after* word.

$$
\begin{aligned}
\langle\text{Sent}\rangle &\models \langle\text{Sent1}\rangle \mid \langle\text{Sent1}\rangle \text{ ',' then } \langle\text{Sent1}\rangle \mid \langle\text{Sent1}\rangle \text{ after you } \langle\text{Sent1}\rangle \\
\langle\text{Sent1}\rangle &\models \langle\text{Clause}\rangle \mid \langle\text{Clause}\rangle \text{ and } \langle\text{Clause}\rangle \\
\langle\text{Clause}\rangle &\models \text{go to } \langle\text{Descr}\rangle \mid \text{pick up } \langle\text{DescrNotDoor}\rangle \mid \text{open } \langle\text{DescrDoor}\rangle \mid \\
&\quad\ \text{put } \langle\text{DescrNotDoor}\rangle \text{ next to } \langle\text{Descr}\rangle \\
\langle\text{DescrDoor}\rangle &\models \langle\text{Article}\rangle \langle\text{Color}\rangle \text{ door } \langle\text{LocSpec}\rangle \\
\langle\text{DescrBall}\rangle &\models \langle\text{Article}\rangle \langle\text{Color}\rangle \text{ ball } \langle\text{LocSpec}\rangle \\
\langle\text{DescrBox}\rangle &\models \langle\text{Article}\rangle \langle\text{Color}\rangle \text{ box } \langle\text{LocSpec}\rangle \\
\langle\text{DescrKey}\rangle &\models \langle\text{Article}\rangle \langle\text{Color}\rangle \text{ key } \langle\text{LocSpec}\rangle \\
\langle\text{Descr}\rangle &\models \langle\text{DescrDoor}\rangle \mid \langle\text{DescrBall}\rangle \mid \langle\text{DescrBox}\rangle \mid \langle\text{DescrKey}\rangle \\
\langle\text{DescrNotDoor}\rangle &\models \langle\text{DescrBall}\rangle \mid \langle\text{DescrBox}\rangle \mid \langle\text{DescrKey}\rangle \\
\langle\text{LocSpec}\rangle &\models \epsilon \mid \text{on your left} \mid \text{on your right} \mid \text{in front of you} \mid \text{behind you} \\
\langle\text{Color}\rangle &\models \epsilon \mid \text{red} \mid \text{green} \mid \text{blue} \mid \text{purple} \mid \text{yellow} \mid \text{grey} \\
\langle\text{Article}\rangle &\models \text{the} \mid \text{a}
\end{aligned}
$$

Figure 2: BNF grammar productions for the Baby Language

The BabyAI platform includes a *verifier* which checks if an agent's sequence of actions successfully achieves the goal of a given instruction within an environment. Descriptors in the language refer to one or to multiple objects. For instance, if an agent is instructed to "go to a red door", it can successfully execute this instruction by going to any of the red doors in the environment. The *then* and *after* connectors can be used to sequence subgoals. The *and* form implies that both subgoals must be completed, without ordering constraints. Importantly, Baby Language instructions leave

go to the red ball

open the door on your left

put a ball next to the blue door

open the yellow door and go to the key behind you

put a ball next to a purple door after you put a blue box next to a grey
box and pick up the purple box

Figure 3: Example Baby Language instructions

details about the execution implicit. An agent may have to find a key and unlock a door, or move obstacles out of the way to complete instructions, without this being stated explicitly.

## 3.3 BABYAI LEVELS

There is abundant evidence in prior literature which shows that a curriculum may greatly facilitate learning of complex tasks for neural architectures (Bengio et al., 2009; Kumar et al., 2010; Zaremba and Sutskever, 2015; Graves et al., 2016). To investigate how a curriculum improves sample efficiency, we created 19 *levels* which require understanding only a limited subset of Baby Language within environments of varying complexity. Formally, a level is a distribution of *missions*, where a mission combines an instruction within an initial environment state. We built levels by selecting *competencies* necessary for each level and implementing a generator to generate missions solvable by an agent possessing only these competencies. Each competency is informally defined by specifying what an agent should be able to do:

- **Room Navigation (ROOM):** navigate a 6x6 room.
- **Ignoring Distracting Boxes (DISTR-BOX):** navigate the environment even when there are multiple distracting grey box objects in it.
- **Ignoring Distractors (DISTR):** same as DISTR-BOX, but distractor objects can be boxes, keys or balls of any color.
- **Maze Navigation (MAZE):** navigate a 3x3 maze of 6x6 rooms, randomly inter-connected by doors.
- **Unblocking the Way (UNBLOCK):** navigate the environment even when it requires moving objects out of the way.
- **Unlocking Doors (UNLOCK):** to be able to find the key and unlock the door if the instruction requires this explicitly.
- **Guessing to Unlock Doors (IMP-UNLOCK):** to solve levels that require unlocking a door, even if this is not explicitly stated in the instruction.
- **Go To Instructions (GOTO):** understand "go to" instructions, e.g. "go to the red ball".
- **Open Instructions (OPEN):** understand "open" instructions, e.g. "open the door on your left".
- **Pickup Instructions (PICKUP):** understand "pick up" instructions, e.g. "pick up a box".
- **Put Instructions (PUT):** understand "put" instructions, e.g. "put a ball next to the blue key".
- **Location Language (LOC):** understand instructions where objects are referred to by relative location as well as their shape and color, e.g. "go to the red ball in front of you".
- **Sequences of Commands (SEQ):** understand composite instructions requiring an agent to execute a sequence of instruction clauses, e.g. "put red ball next to the green box after you open the door".

Table 1 lists all current BabyAI levels together with the competencies required to solve them. These levels form a progression in terms of the competencies required to solve them, culminating with

Table 1: BabyAI Levels and the required competencies

| | ROOM | DISTR-BOX | DISTR | MAZE | UNBLOCK | UNLOCK | IMP-UNLOCK | GOTO | OPEN | PICKUP | PUT | LOC | SEQ |
|---|---|---|---|---|---|---|---|---|---|---|---|---|---|
| GoToObj | x | | | | | | | | | | | | |
| GoToRedBallGrey | x | x | | | | | | | | | | | |
| GoToRedBall | x | x | x | | | | | | | | | | |
| GoToLocal | x | x | x | | | | | x | | | | | |
| PutNextLocal | x | x | x | | | | | | | | x | | |
| PickupLoc | x | x | x | | | | | | | x | | x | |
| GoToObjMaze | x | | | x | | | | | | | | | |
| GoTo | x | x | x | x | | | | x | | | | | |
| Pickup | x | x | x | x | | | | | | x | | | |
| UnblockPickup | x | x | x | x | x | | | | | x | | | |
| Open | x | x | x | x | | | | | x | | | | |
| Unlock | x | x | x | x | | x | | | x | | | | |
| PutNext | x | x | x | x | | | | | | | x | | |
| Synth | x | x | x | x | x | x | | x | x | x | x | | |
| SynthLoc | x | x | x | x | x | x | | x | x | x | x | x | |
| GoToSeq | x | x | x | x | | | | x | | | | | x |
| SynthSeq | x | x | x | x | x | x | | x | x | x | x | x | x |
| GoToImpUnlock | x | x | x | x | | | x | x | | | | | |
| BossLevel | x | x | x | x | x | x | x | x | x | x | x | x | x |

the BossLevel, which requires mastering all competencies. The definitions of competencies are informal and should be understood in the minimalistic sense, i.e. to test the ROOM competency we have built the GoToObj level where the agent needs to reach the only object in an empty room. Note that the GoToObj level does not require the GOTO competency, as this level can be solved without any language understanding, since there is only a single object in the room. However, solving the GoToLocal level, which instructs the agent to go to a specific object in the presence of multiple distractors, requires understanding GOTO instructions.

## 3.4 THE BOT AGENT

The bot is a key ingredient intended to perform the role of a simulated human teacher. For any of the BabyAI levels, it can generate demonstrations or suggest actions for a given environment state. Whereas the BabyAI learner is meant to be generic and should scale to new and more complex tasks, the bot is engineered using knowledge of the tasks. This makes sense since the bot stands for the human in the loop, who is supposed to understand the environment, how to solve missions, and how to teach the baby learner. The bot has direct access to a tree representation of instructions, and so does not need to parse the Baby Language. Internally, it executes a stack machine in which instructions and subgoals are represented (more details can be found in Appendix C). The stack-based design allows the bot to interrupt what it is currently doing to achieve a new subgoal, and then resume the original task. For example, going to a given object will require exploring the environment to find that object.

The subgoals which the bot implements are:

- **Open:** Open a door that is in front of the agent.
- **Close:** Close a door that is in front of the agent.
- **Pickup:** Execute the pickup action (pick up an object).
- **Drop:** Execute the drop action (drop an object being carried).
- **GoNextTo:** Go next to an object matching a given (type, color) description or next to a cell at a given position.

- **Explore:** Uncover previously unseen parts of the environment.

All of the Baby Language instructions are decomposed into these internal subgoals which the bot knows how to solve. Many of these subgoals, during their execution, can also push new subgoals on the stack. A central part of the design of the bot is that it keeps track of the grid cells of the environment which it has and has not seen. This is crucial to ensure that the bot can only use information which it could realistically have access to by exploring the environment. Exploration is implemented as part of the Explore subgoal, which is recursive. For instance, exploring the environment may require opening doors, or moving objects that are in the way. Opening locked doors may in turn require finding a key, which may itself require exploration and moving obstructing objects. Another key component of the bot's design is a shortest path search routine. This is used to navigate to objects, to locate the closest door, or to navigate to the closest unexplored cell.

## 4 Experiments

We assess the difficulty of BabyAI levels by training a behavioral cloning baseline for each level. Furthermore, we estimate how much data is required to solve some of the simpler levels and study to which extent the data demands can be reduced by using basic curriculum learning and interactive teaching methods. All the code that we use for the experiments, as well as containerized pretrained models, is available online.

### 4.1 Setup

The BabyAI platform provides by default a 7x7x3 symbolic observation $x_t$ (a partial and local egocentric view of the state of the environment) and a variable length instruction $c$ as inputs at each time step. We use a basic model consisting of standard components to predict the next action $a$ based on $x$ and $c$. In particular, we use a GRU (Cho et al., 2014) to encode the instruction and a convolutional network with two batch-normalized (Ioffe and Szegedy, 2015) FiLM (Perez et al., 2017) layers to jointly process the observation and the instruction. An LSTM (Hochreiter and Schmidhuber, 1997) memory is used to integrate representations produced by the FiLM module at each step. Our model is thus similar to the gated-attention model used by Chaplot et al. (2018), inasmuch as gated attention is equivalent to using FiLM without biases and only at the output layer.

We have used two versions of our model, to which we will refer as the Large model and the Small model. In the Large model, the memory LSTM has 2048 units and the instruction GRU is bidirectional and has 256 units. Furthermore, an attention mechanism (Bahdanau et al., 2015) is used to focus on the relevant states of the GRU. The Small model uses a smaller memory of 128 units and encodes the instruction with a unidirectional GRU and no attention mechanism.

In all our experiments, we used the Adam optimizer (Kingma and Ba, 2015) with the hyperparameters $\alpha = 10^{-4}$, $\beta_1 = 0.9$, $\beta_2 = 0.999$ and $\epsilon = 10^{-5}$. In our imitation learning (IL) experiments, we truncated the backpropagation through time at 20 steps for the Small model and at 80 steps for the Large model. For our reinforcement learning experiments, we used the Proximal Policy Optimization (PPO, Schulman et al., 2017) algorithm with parallelized data collection. Namely, we performed 4 epochs of PPO using 64 rollouts of length 40 collected with multiple processes. We gave a non-zero reward to the agent only when it fully completed the mission, and the magnitude of the reward was $1 - 0.9n/n_{max}$, where $n$ is the length of the successful episode and $n_{max}$ is the maximum number of steps that we allowed for completing the episode, different for each mission. The future returns were discounted with a factor $\gamma = 0.99$. For generalized advantage estimation (Schulman et al., 2015) in PPO we used $\lambda = 0.99$.

In all our experiments we reported the success rate, defined as the ratio of missions of the level that the agent was able to accomplish within $n_{max}$ steps.

Running the experiments outlined in this section required between 20 and 50 GPUs over two weeks. At least as much computing was required for preliminary investigations.

Table 2: Baseline imitation learning results for all BabyAI levels. Each model was trained with 1M demonstrations from the respective level. For reference, we also list the mean and standard deviation of demonstration length for each level.

| Model | Success Rate (%) | Demo Length (Mean $\pm$ Std) |
|---|---|---|
| GoToObj | 100 | 5.18$\pm$2.38 |
| GoToRedBallGrey | 100 | 5.81$\pm$3.29 |
| GoToRedBall | 100 | 5.38$\pm$3.13 |
| GoToLocal | 99.8 | 5.04$\pm$2.76 |
| PutNextLocal | 99.2 | 12.4$\pm$4.54 |
| PickupLoc | 99.4 | 6.13$\pm$2.97 |
| GoToObjMaze | 99.9 | 70.8$\pm$48.9 |
| GoTo | 99.4 | 56.8$\pm$46.7 |
| Pickup | 99 | 57.8$\pm$46.7 |
| UnblockPickup | 99 | 57.2$\pm$50 |
| Open | 100 | 31.5$\pm$30.5 |
| Unlock | 98.4 | 81.6$\pm$61.1 |
| PutNext | 98.8 | 89.9$\pm$49.6 |
| Synth | 97.3 | 50.4$\pm$49.3 |
| SynthLoc | 97.9 | 47.9$\pm$47.9 |
| GoToSeq | 95.4 | 72.7$\pm$52.2 |
| SynthSeq | 87.7 | 81.8$\pm$61.3 |
| GoToImpUnlock | 87.2 | 110$\pm$81.9 |
| BossLevel | 77 | 84.3$\pm$64.5 |

## 4.2 BASELINE RESULTS

To obtain baseline results for all BabyAI levels, we have trained the Large model (see Section 4.1) with imitation learning using one million demonstration episodes for each level. The demonstrations were generated using the bot described in Section 3.4. The models were trained for 40 epochs on levels with a single room and for 20 epochs on levels with a 3x3 maze of rooms. Table 2 reports the maximum success rate on a validation set of 512 episodes. All of the single-room levels are solved with a success rate of 100.0%. As a general rule, levels for which demonstrations are longer tend to be more difficult to solve.

Using 1M demonstrations for levels as simple as GoToRedBall is very inefficient and hardly ever compatible with the long-term goal of enabling human teaching. The BabyAI platform is meant to support studies of how neural agents can learn with less data. To bootstrap such studies, we have computed baseline sample efficiencies for imitation learning and reinforcement learning approaches to solving BabyAI levels. We say an agent solves a level if it reaches a success rate of at least 99%. We define the sample efficiency as the minimum number of demonstrations or RL episodes required to train an agent to solve a given level. To estimate the thus defined sample efficiency for imitation learning while staying within a reasonable computing budget, we adopt the following procedure. For a given level, we first run three experiments with $10^6$ demonstrations. In the remaining $M$ experiments we use $k_1 = 2^{l_0}, k_2 = 2^{l_0+d}, \ldots, k_M = 2^{l_0+(M-1)d}$ demonstrations respectively. We use different values of $l_0$, $M$ for each level to ensure that we run experiments with not enough, just enough and more than enough demonstrations. Same value of $d = 0.2$ is used in all imitation learning experiments. For each experiment $i$, we measure the best smoothed online validation performance $s_i$ that is achieved during the first $2T$ training steps, where $T = (T_1 + T_2 + T_3)/3$ is the average number of training steps required to solve the level in the three runs with $10^6$ demonstrations. We then fit a Gaussian Process (GP) model (Rasmussen and Williams, 2005) with noisy observations using $(k_i, s_i)$ as training data in order to interpolate between these data points. The GP posterior is fully tractable, which allows us to compute analytically the posterior distribution of the expected success rate, as well as the posterior over the minimum number of samples $k_{min}$ that is sufficient to solve the level. We report the 99% credible interval for $k_{min}$. We refer the reader to Appendix A for a more detailed explanation of this procedure.

We estimate sample efficiency of imitation learning on 6 chosen levels. The results are shown in Table 3 (see "IL from Bot" column). In the same table (column "RL") we report the 99% confidence

Table 3: The sample efficiency of imitation learning (IL) and reinforcement learning (RL) as the number of demonstrations (episodes) required to solve each level. All numbers are thousands. For the imitation learning results we report a 99% credible interval. For RL experiments we report the 99% confidence interval. See Section 4 for details.

| Level | IL from Bot | RL |
|---|---|---|
| GoToRedBallGrey | 8.431 - 12.43 | 15.9 - 17.4 |
| GoToRedBall | 49.67 - 62.01 | 261.1 - 333.6 |
| GoToLocal | 148.5 - 193.2 | 903 - 1114 |
| PickupLoc | 204.3 - 241.2 | 1447 - 1643 |
| PutNextLocal | 244.6 - 322.7 | 2186 - 2727 |
| GoTo | 341.1 - 408.5 | 816 - 1964 |

Table 4: The sample efficiency results for pretraining experiments. For each pair of base levels and target levels that we have tried, we report how many demonstrations (in thousands) were required, as well as the baseline number of demonstrations required for training from scratch. In both cases we report a 99% credible interval, see Section 4 for details. Note how choosing the right base levels (e.g. GoToLocal instead of GoToObjMaze) is crucial for pretraining to be helpful.

| Base Levels | Target Level | Without Pretraining | With Pretraining |
|---|---|---|---|
| GoToLocal | GoTo | 341 - 409 | 183 - 216 |
| GoToObjMaze | GoTo | 341 - 409 | 444 - 602 |
| GoToLocal-GoToObjMaze | GoTo | 341 - 409 | 173 - 216 |
| GoToLocal | PickupLoc | 204 - 241 | 71.2 - 88.9 |
| GoToLocal | PutNextLocal | 245 - 323 | 188 - 231 |

interval for the number of episodes that were required to solve each of these levels with RL, and as expected, the sample efficiency of RL is substantially worse than that of IL (anywhere between 2 to 10 times in these experiments).

To analyze how much the sample efficiency of IL depends on the source of demonstrations, we try generating demonstrations from agents that were trained with RL in the previous experiments. The results for the 3 easiest levels are reported in the "IL from RL Expert" column in Table 5. Interestingly, we found that the demonstrations produced by the RL agent are easier for the learner to imitate. The difference is most significant for GoToRedBallGrey, where less than 2K and more than 8K RL and bot demonstrations respectively are required to solve the level. For GoToRedBall and GoToLocal, using RL demonstrations results in 1.5-2 times better sample efficiency. This can be explained by the fact that the RL expert has the same neural network architecture as the learner.

### 4.3 CURRICULUM LEARNING

To demonstrate how curriculum learning research can be done using the BabyAI platform, we perform a number of basic pretraining experiments. In particular, we select 5 combinations of base levels and a target level and study whether pretraining on base levels can help the agent master the target level with fewer demonstrations. The results are reported in Table 4. In four cases, using GoToLocal as one of the base levels reduces the number of demonstrations required to solve the target level. However, when only GoToObjMaze was used as the base level, we have not found pretraining to be beneficial. We find this counter-intuitive result interesting, as it shows how current deep learning methods often can not take the full advantage of available curriculums.

### 4.4 INTERACTIVE LEARNING

Lastly, we perform a simple case study of how sample efficiency can be improved by interactively providing more informative examples to the agent based on what it has already learned. We experiment with an iterative algorithm for adaptively growing the agent's training set. In particular, we start with $2^{10}$ base demonstrations, and at each iteration we increase the dataset size by a factor of $2^{1/4}$ by providing bot demonstrations for missions on which the agent failed. After each dataset increase we train a new agent from scratch. We perform such dataset increases until the dataset

Table 5: The sample efficiency of imitation learning (IL) from an RL-pretrained expert and interactive imitation learning defined as the number of demonstrations required to solve each level. All numbers are in thousands. 99% credible intervals are reported in all experiments, see Section 4 for details.

| Level | IL from Bot | IL from RL Expert | Interactive IL from Bot |
|---|---|---|---|
| GoToRedBallGrey | 8.43 - 12.4 | 1.53 - 2.11 | 1.71 - 1.88 |
| GoToRedBall | 49.7 - 62 | 36.6 - 44.5 | 31.8 - 36 |
| GoToLocal | 148 - 193 | 74.2 - 81.8 | 93 - 107 |

reaches the final size is clearly sufficient to achieve 99% success rate. We repeat the experiment 3 times for levels GoToRedBallGrey, GoToRedBall and GoToLocal and then estimate how many interactively provided demonstrations would be required for the agent be 99% successful for each of these levels. To this end, we use the same GP posterior analysis as for regular imitation learning experiments.

The results for the interactive imitation learning protocol are reported in Table 5. For all 3 levels that we experimented with, we have observed substantial improvement over the vanilla IL, which is most significant (4 times less demonstrations) for GoToRedBallGrey and smaller (1.5-2 times less demonstrations) for the other two levels.

## 5 CONCLUSION & FUTURE WORK

We present the BabyAI research platform to study language learning with a human in the loop. The platform includes 19 levels of increasing difficulty, based on a decomposition of tasks into a set of basic competencies. Solving the levels requires understanding the Baby Language, a subset of English with a formally defined grammar which exhibits compositional properties. The language is minimalistic and the levels seem simple, but empirically we have found them quite challenging to solve. The platform is open source and extensible, meaning new levels and language concepts can be integrated easily.

The results in Section 4 suggest that current imitation learning and reinforcement learning methods scale and generalize poorly when it comes to learning tasks with a compositional structure. Hundreds of thousands of demonstrations are needed to learn tasks which seem trivial by human standards. Methods such as curriculum learning and interactive learning can provide measurable improvements in terms of sample efficiency, but, in order for learning with an actual human in the loop to become realistic, an improvement of at least three orders of magnitude is required.

An obvious direction of future research to find strategies to improve sample efficiency of language learning. Tackling this challenge will likely require new models and new teaching methods. Approaches that involve an explicit notion of modularity and subroutines, such as Neural Module Networks (Andreas et al., 2016) or Neural Programmer-Interpreters (Reed and de Freitas, 2015), seem like a promising direction. It is our hope that the BabyAI platform can serve as a challenge and a benchmark for the sample efficiency of language learning for years to come.

## ACKNOWLEDGEMENTS

We thank Tristan Deleu, Saizheng Zhang for useful discussions. We also thank Rachel Samson, Léonard Boussioux and David Yu-Tung Hui for their help in preparing the final version of the paper. This research was enabled in part by support provided by Compute Canada (www.computecanada.ca), NSERC and Canada Research Chairs. We also thank Nvidia for donating NVIDIA DGX-1 used for this research.

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

## A  SAMPLE EFFICIENCY ESTIMATION

### A.1  REINFORCEMENT LEARNING

To estimate the number of episodes required for an RL agent to solve a BabyAI level, we monitored the agent's smoothed online success rate. We recorded the number of training episodes after which the smoothed performance crossed the 99% success rate threshold. Each experiment was repeated 10 times and the 99% t-test confidence interval is reported in Table 3.

### A.2  IMITATION LEARNING

Estimating how many demonstrations is required for imitation learning to achieve a given performance level is challenging. In principle, one can sample a dense grid of dataset sizes, train the model until full convergence on each of the resulting datasets, and find the smallest dataset size for which on average the model's best performance exceeds the target level. In practice, such a procedure would be prohibitively computationally expensive.

To make sample efficiency estimation practical, we designed a relatively cheap semi-automatic approximate protocol. We minimize computational resources by using early-stopping and non-parametric interpolation between different data points.

**Early Stopping Using Normal Time**  Understanding if a training run has converged and if the model's performance will not improve any further is non-trivial. To early-stop models in a consistent automatic way, we estimate the "normal" time $T$ that training a model on a given level would take if an unlimited (in our case $10^6$) number of demonstrations was available. To this end, we train 3 models with $10^6$ demonstrations. We evaluate the online success rate after every 100 or 400 (depending on the model size) batches, each time using 512 different episodes. The online success rate is smoothed using a sliding window of length 10. Let $s(k, j, t)$ denote the smoothed online performance for the $j$-th run with $k$ demonstrations at time $t$. Using this notation, we compute the normal time $T$ as $T = (T_1 + T_2 + T_3)/3$, where $T_i = \min_t \left\{ t : s_j(10^6, j, t) > 99 \right\}$. Once $T$ is computed, it is used to early stop the remaining $M$ runs that use different numbers of demonstrations $k_i$. Namely the result $s_i$ of the $i$-th of these runs is computed as $s_i = \max_{t < 2T} s(k_i, 1, t)$.

**Interpolation Using Gaussian Processes**  Given the success rate measurements $D = \{(k_i, s_i)\}_{i=1}^M$, $k_1 < k_2 < \ldots < k_M$, we estimate the minimum number of samples $k_{min}$ that is required for the model to reach 99% average success rate. To this end, we a Gaussian Process (GP) model to interpolate between the available $(k_i, s_i)$ data points (Rasmussen and Williams, 2005). GP is a popular model for non-linear regression, whose main advantage is principled modelling of predictions' uncertainty.

Specifically, we model the dependency between the success rate $s$ and the number of examples $k$ as follows:

$$f \sim GP_{RBF}(l), \tag{1}$$
$$\tilde{s}(k) = 99 + \sigma_f f(\log_2 k), \tag{2}$$
$$\epsilon(k) \sim N(0, 1), \tag{3}$$
$$s(k) = \tilde{s}(k) + \sigma_\epsilon \epsilon(k), \tag{4}$$

where $RBF$ reflects the fact that we use the Radial Basis Function kernel, $l$ is the kernel's length-scale parameter, $\epsilon(k)$ is white noise, $\sigma_f$ and $\sigma_\epsilon$ add scaling to the GP $f$ and the noise $\epsilon$. Note the distinction between the average and the observed performances $\tilde{s}(k)$ and $s(k)$. Using the introduced notation, $k_{min}$ can be formally defined as $k_{min} = \min_{k \in [k_1; k_M]} \tilde{s}(k) = 99$.

To focus on the interpolation in the region of interest, we drop all $(k_i, s_i)$ data points for which $s_i < 95$. We then fit the model's hyperparameters $l$, $\sigma_f$ and $\sigma_\epsilon$ by maximizing the likelihood of the remaining data points. To this end, we use the implementation from scikit-learn (Pedregosa et al., 2011). Once the model is fit, it defines a Gaussian posterior density $p(\tilde{s}(k_1'), \ldots, \tilde{s}(k_{M'}')|D)$ for any $M'$ data points $k_1', k_2', \ldots, k_{M'}'$. It also defines a probability distribution $p(k_{min}|D)$. We are not

aware of an analytic expression for $p(k_{min}|D)$, and hence we compute a numerical approximation as follows. We sample a dense log-scale grid of $M'$ points $k'_1, k'_2, \ldots, k'_{M'}$ in the range $[k_1; k_M]$. For each number of demonstrations $k'_i$ we approximate the probability $p(k'_{i-1} < k_{min} < k'_i|D)$ that $\tilde{s}(k)$ crosses the 99% threshold somewhere between $k'_{i-1}$ and $k'_i$ as follows:

$$p(k'_{i-1} < k_{min} < k'_i|D) \approx p'_i = p(\tilde{s}(k'_1) < 99, \ldots, \tilde{s}(k'_{i-1}) < 99, \tilde{s}(k'_i) > 99|D) \qquad (5)$$

Equation 5 is an approximation because the posterior $\tilde{s}$ is not necessarily monotonic. In practice, we observed that the monotonic nature of the observed data $D$ shapes the posterior accordingly. We use the probabilities $p'_i$ to construct the following discrete approximation of the posterior $p(k_{min}|D)$:

$$p(k_{min}|D) \approx \sum_{i=1}^{M} p'_i \delta(k'_i) \qquad (6)$$

where $\delta(k'_i)$ are Dirac delta-functions. Such a discrete approximation is sufficient for the purpose of computing 99% credible intervals for $k_{min}$ that we report in the paper.

# B  MINIGRID ENVIRONMENTS FOR OPENAI GYM

The environments used for this research are built on top of MiniGrid, which is an open source grid-world package. This package includes a family of reinforcement learning environments compatible with the OpenAI Gym framework. Many of these environments are parameterizable so that the difficulty of tasks can be adjusted (e.g. the size of rooms is often adjustable).

## B.1  THE WORLD

In MiniGrid, the world is a grid of size NxN. Each tile in the grid contains exactly zero or one object, and the agent can only be on an empty tile or on a tile containing an open door. The possible object types are wall, door, key, ball, box and goal. Each object has an associated discrete color, which can be one of red, green, blue, purple, yellow and grey. By default, walls are always grey and goal squares are always green.

## B.2  REWARD FUNCTION

Rewards are sparse for all MiniGrid environments. Each environment has an associated time step limit. The agent receives a positive reward if it succeeds in satisfying an environment's success criterion within the time step limit, otherwise zero. The formula for calculating positive sparse rewards is $1 - 0.9 * (step\_count/max\_steps)$. That is, rewards are always between zero and one, and the quicker the agent can successfully complete an episode, the closer to 1 the reward will be. The $max\_steps$ parameter is different for each mission, and varies depending on the size of the environment (larger environments having a higher time step limit) and the length of the instruction (more time steps are allowed for longer instructions).

## B.3  ACTION SPACE

There are seven actions in MiniGrid: turn left, turn right, move forward, pick up an object, drop an object, toggle and done. The agent can use the turn left and turn right action to rotate and face one of 4 possible directions (north, south, east, west). The move forward action makes the agent move from its current tile onto the tile in the direction it is currently facing, provided there is nothing on that tile, or that the tile contains an open door. The agent can open doors if they are right in front of it by using the toggle action.

## B.4  OBSERVATION SPACE

Observations in MiniGrid are partial and egocentric. By default, the agent sees a square of 7x7 tiles in the direction it is facing. These include the tile the agent is standing on. The agent cannot see through walls or closed doors. The observations are provided as a tensor of shape 7x7x3. However, note that these are not RGB images. Each tile is encoded using 3 integer values: one describing the

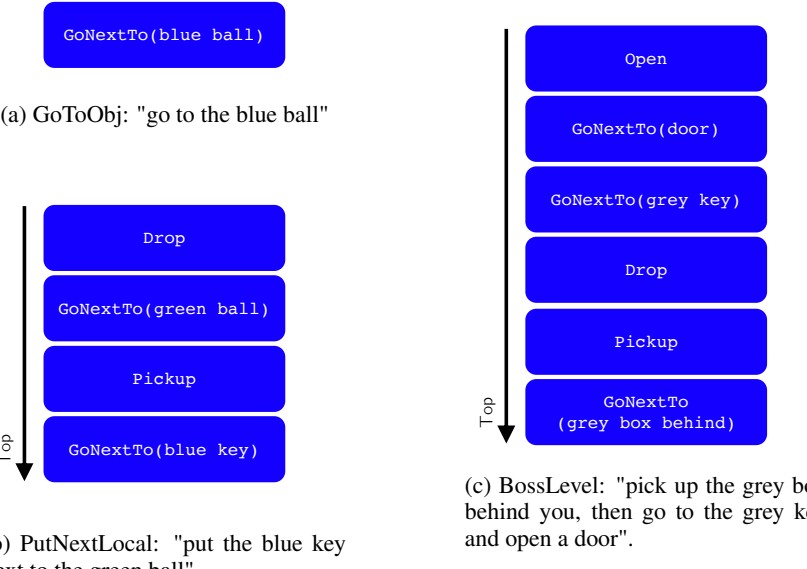

(a) GoToObj: "go to the blue ball"

(b) PutNextLocal: "put the blue key next to the green ball"

(c) BossLevel: "pick up the grey box behind you, then go to the grey key and open a door".

Figure 4: Examples of initial stacks corresponding to three different instructions.

type of object contained in the cell, one describing its color, and a state indicating whether doors are open, closed or locked. This compact encoding was chosen for space efficiency and to enable faster training. The fully observable RGB image view of the environments shown in this paper is provided for human viewing.

## C  BOT IMPLEMENTATION DETAILS

### C.1  TRANSLATION OF INSTRUCTIONS INTO SUBGOALS

The bot has access to a representation of the instructions for each environment. These instructions are decomposed into subgoals that are added to a stack. In Figure 4 we show the stacks corresponding to the examples in Figure 1. The stacks are illustrated in bottom to top order, that is, the lowest subgoal in the illustration is to be executed first.

### C.2  PROCESSING OF SUBGOALS

Once instructions for a task are translated into the initial stack of subgoals, the bot starts by processing the first subgoal. Each subgoal is processed independently, and can either lead to more subgoals being added to the stack, or to an action being taken. When an action is taken, the state of the bot in the environment changes, and its visibility mask is populated with all the new observed cells and objects, if any. The visibility mask is essential when looking for objects and paths towards cells, because it keeps track of what the bot has seen so far. Once a subgoal is marked as completed, it is removed from the stack, and the bot starts processing the next subgoal in the stack. Note that the same subgoal can remain on top of the stack for multiple time steps, and result in multiple actions being taken.

The `Close`, `Drop` and `Pickup` subgoals are trivial, that is, they result in the execution of the corresponding action and then immediately remove themselves from the stack. Diagrams depicting how the `Open`, `GoNextTo` and `Explore` subgoals are handled are depicted in Figures 5, 6, and 7 respectively. In the diagrams, we use the term "forward cell" to refer to the grid cell that the agent is facing. We say that a path from X to Y contains blockers if there are objects that need to be moved in order for the agent to be able to navigate from X to Y. A "clear path" is a path without blockers.

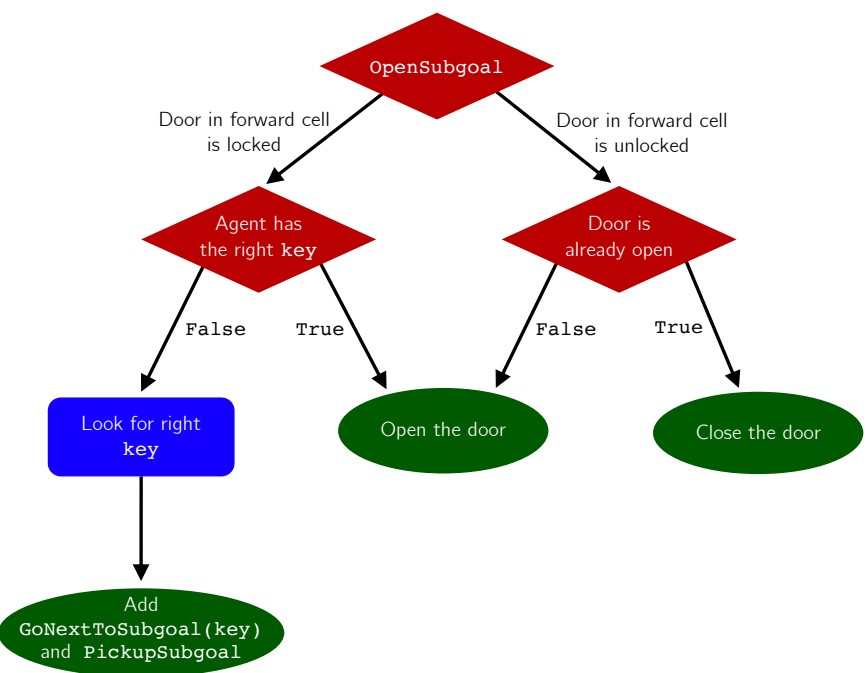

Figure 5: Processing of the Open subgoal

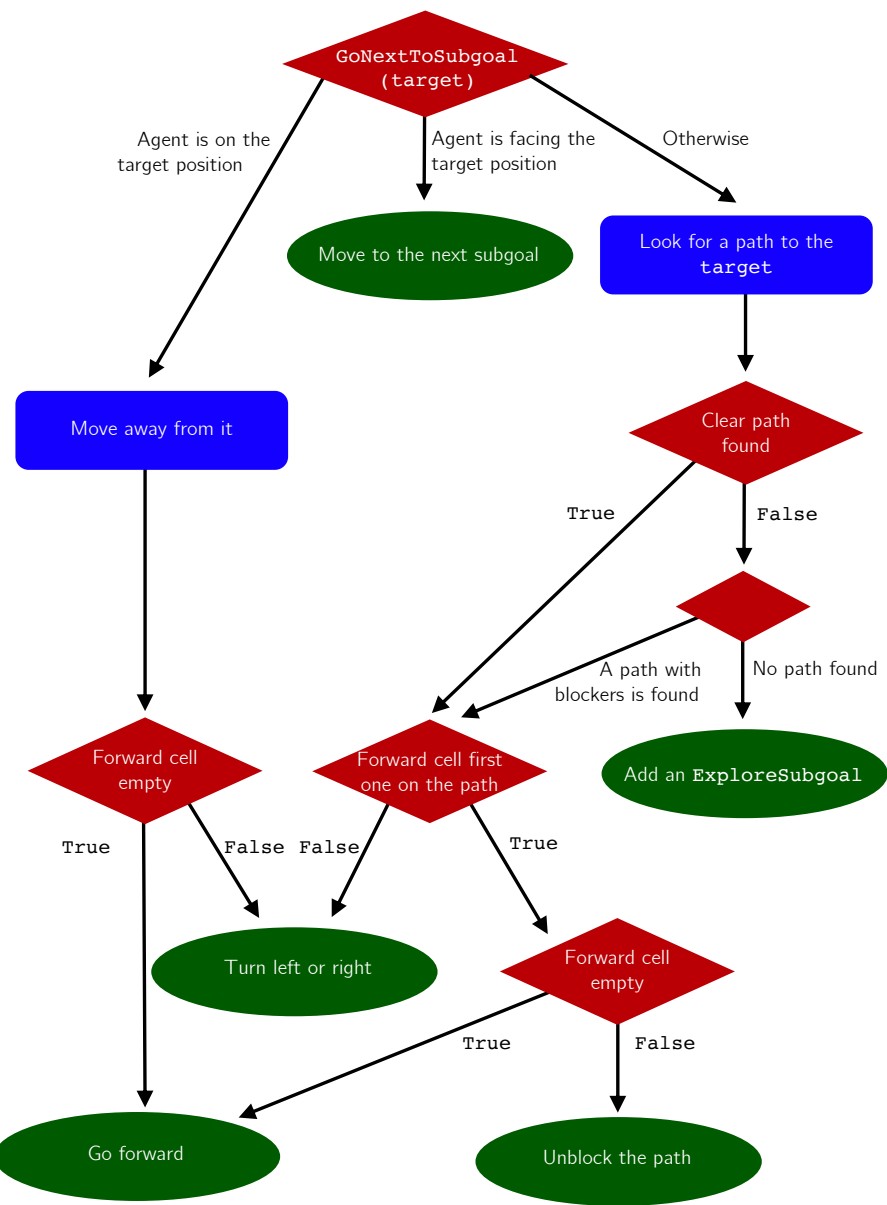

Figure 6: Processing of the GoNextTo subgoal

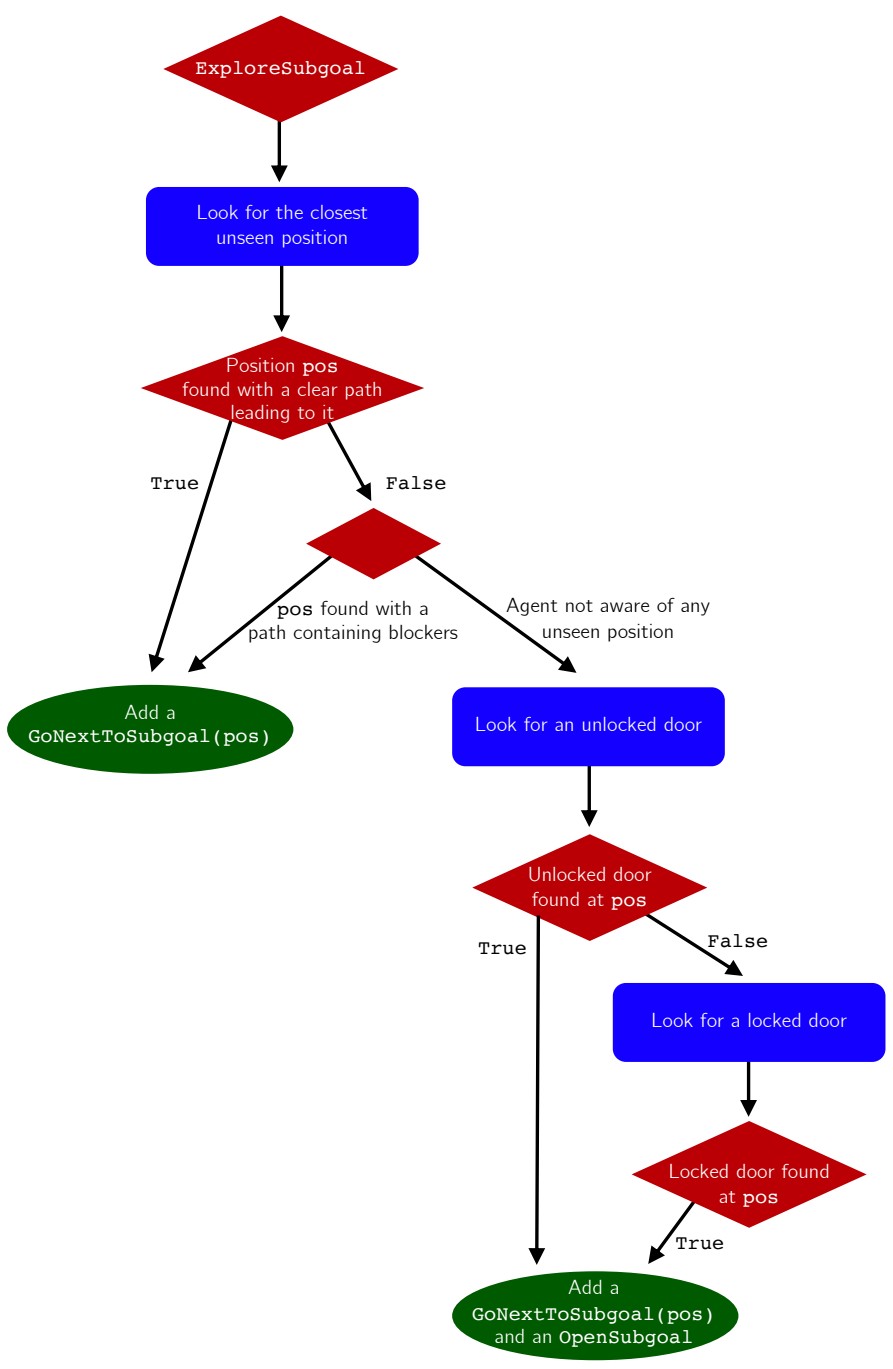

Figure 7: Processing of the Explore subgoal

