# OpenReview forum: "BabyAI: A Platform to Study the Sample Efficiency of Grounded Language Learning"
_ICLR.cc/2019/Conference_

### Official Review · AnonReviewer3 · 2018-10-27
**Interesting direction and open-source platform, but paper falls short of human evaluation**

**Rating:** 6
**Confidence:** 4

**Review:**

Summary

The authors introduce BabyAI, a platform with the aim to study grounded language learning with a human in the loop. The platform includes a *simulated* human expert (bot) that teaches a neural learner. The current domain used in a 2D gridworld and the synthetic instructions require the agent to navigate the world (including unlocking doors) and move objects to specified locations. They also introduce "Baby Language" to give instructions to the agent as well as to automatically verify their execution.

The paper includes a detailed description of the minigrid env with the included tasks and instruction language set.

Authors trained small and large LSTM models on the tasks on a variety of standard learning approaches, using pure exploration (RL) and imitation from a synthetic bot (IL). They show IL is much more data efficient than RL in this domain as well. Also, a curriculum approach is evaluated (pre-train on task 1-N, then train on task N+1).

Pro
- Human-in-the-loop research is an exciting direction.
- The language instruction set is a starting point for high-level human instructions.

Con
- It is still unclear how to effectively learn with human-in-the-loop. The authors don't actually evaluate
1) how well the bot imitates a human, or
2) how an actual human would interact and speed up learning.
All experiments are done with standard learning approaches with a synthetic bot.
- The authors assume that human feedback comes as instructions or demonstrations. These are not the only forms of feedback possible (e.g., preferences). (Does the platform easily support those?)

Reproducibility
- Open-sourcing the platform is a good contribution to the community.

---

> ### Author Response · Authors · 2018-11-15
> **Response to Reviewer 3**
>
> We thank Reviewer 3 (R3) for their review that nicely summarizes our paper. In our response we discuss the two main questions that R3 asked, namely why we have not yet performed experiments with an actual human and how other learning approaches can be studied using the BabyAI platform.
>
> We completely agree with R3 that “it is still unclear how to effectively learn with a human in the loop.” Our baselines studies on the BabyAI platform have indeed shown that standard approaches for grounded language understanding would need from thousands to hundreds of thousands of demonstrations by the human to learn even the simplest tasks. In order to proceed to studies with actual humans in the loop, the human should be able to see in real time how the agent is progressing, and with the approaches that we evaluated such progress would be unbearably slow. We have postponed studies with actual humans until a sufficient progress is made on BabyAI, that is until we have levels that (at least with pretraining) can be mastered with hundreds of demonstrations. For now, we believe the BabyAI platform is already useful as it supports rigorous studies on data efficiency of grounded language understanding, so that we can measure progress towards the goal of learning with a human in the loop .
>
> The reviewer correctly pointed out that there are other ways in which a human could teach an agent for which we do not provide baseline results, in particular learning with preferences. To the best of our understanding studies of learning with preferences could be done using the BabyAI platform, since human preferences could be simulated by using the instruction verifier. We would however expect the data efficiency of learning with preferences to be closer to that of RL (i.e. millions of episodes) than that of imitation learning (hundreds of thousand demonstrations).
>
> We hope that R3 finds the clarifications that we present in this response informative and useful.

---

### Official Review · AnonReviewer2 · 2018-11-03
**New platfrom for research**

**Rating:** 7
**Confidence:** 5

**Review:**

Summary:
This paper presents a research platform with a simulated human (a.k.a bot) in the loop for learning to execute language instructions in which language has compositional structures. The language introduced in this paper can be used to instruct an agent to go to objects, pick up objects, open doors, and put objects next to other objects. MiniGrid is used to build the environments used for this platform. In addition to introducing the platform, they evaluate the difficulty of each level by training an imitation learning baseline using one million demonstration episodes for each level and report results. Moreover, the reported results contain data efficiencies for imitation learning and reinforcement learning based approaches to solving BabyAI levels.

A platform like this can be very useful to expedite research in language learning, machine learning, etc. In my view, work like this should be highly encouraged by this conference and alike.

Comments:
1.  There are following papers should be cited as they are very related to this paper:
    a) Vision-and-Language Navigation: Interpreting visually-grounded navigation instructions in real environments
        https://arxiv.org/abs/1711.07280
    b) AI2-THOR: An Interactive 3D Environment for Visual AI
         https://arxiv.org/abs/1712.05474
2. Paper is well-written and easy to follow. The only part which needs an improvement is section 3.4 as the text is a bit confusing.
3. Heuristic expert, simulated human, human, and bot are exchangeably used in the text. It is better to pick one of these to avoid any confusion in the text. In general, it is not clear to why 'a human in the loop' is chosen, isn't just a program/bot that has is engineered using knowledge of the tasks?
4. In Table 5, in GoToObjMaze row, data efficiency for "With Pretraining" is greater than "Without Pretraining", is this a typo? if not, why this is the case?
5. One useful baseline which can be added to this paper is task-oriented language grounding. This task will be a better measure than current baselines, especially for RL case. Authors can check out the following paper:
Gated-Attention Architectures for Task-Oriented Language Grounding
https://arxiv.org/abs/1706.07230
The code is available for this paper.

Question:
When this platform will be available for public?

---

> ### Author Response · Authors · 2018-11-15
> **Response to Reviewer 2**
>
> We are grateful for the detailed review provided by Reviewer 2 (R2), their careful reading of the paper, and their suggestions. We will do our best to incorporate these.
>
> R2 has asked when the platform will be available to the public, and we are happy to announce that that the platform is already available on Github. We will add a link to the repository when the review process is complete in order to preserve anonymity. We have also published a Docker image with the baseline models on Docker Hub so that our results can be easily replicated.
>
> We respond below to the 5 specific comments that R2 made in their review in same order as they were originally presented:
>
> 1) We will cite the two highly relevant papers that R2 suggested in our Related Work section.
> 2) We will improve the description of the stack-based bot in the text of Section 3.4, and we will furthermore add the complete pseudocode for the bot in an Appendix.
> 3) We will follow R2’s suggestion and be more consistent the choice of terms  “simulated human”, “heuristic expert” and “bot”, in particular we will try to use “bot” as often as possible. We would still like to retain occasional references to human in the loop training, since this it is this kind of training that our research aspires to eventually enable. The bot’s role in this context is simulating a hypothetical human teacher, which is why occasionally the paper uses the words “simulated human”.
> 4) We fully share R2’s concerns with regard to a counterintuitive decrease of data efficiency on GoTo that is observed when the model is first pretrained on GoToObjMaze. We are however confident that this result is correct, and moreover we find it rather interesting. An agent that is pretrained on “GoToObjMaze” level knows how to efficiently navigate a 3x3 maze of 6x6 rooms (Table 1), but somehow this pretraining does not help at all when the same agent is then trained to perform a range of similar tasks (defined by instructions like “go to red ball”) and in the presence of distractor objects. We treat this as an evidence that the current deep learning architectures for language understanding do not lend themselves to curriculum learning, and we hope that BabyAI will support studies on how to improve them in this respect. We believe that such much needed improvements in curriculum learning will play a key role in enabling actual human in the loop training, which is the main aspiration of the paper.
> 5) We thank R2 for the pointer to the “Gated-Attention Architectures for Task-Oriented Language Grounding” paper. We have already cited this paper in Introduction and Related Work. We were not sure which of the two ways of understanding  the author’s suggestion that “This task will be a better measure than current baselines, especially for RL case” is right, and below we comment on both.
>
> If R2’s suggestion was to consider using the VizDoom environment from the aforementioned paper, then we would like to note that we did consider the option of using this environment along with other ones mentioned in our Related Work section. We concluded that in order to have the specific combination of features that we wanted (high speed, interacting with objects in the environment, systematically designed language), we could not use existing environments, such as VizDoom, and had to build a new MiniGrid environment and implement the Baby language in it.
>
> If instead R2 suggested to use the gated attention approach to combine  representations of images and instructions, then we would like to note that the FiLM [1] layers that we use in our model perform a very similar computation (and in fact FiLM and the “Task-Oriented Language Grounding” were both presented at the same AAAI 2018 conference). We will make this connection more clear in the text of the paper where we describe the model that we use in our experiments.
>
> We hope that R2 finds our clarifications and comments helpful.
>
> [1] FiLM: Visual Reasoning with a General Conditioning Layer (https://arxiv.org/abs/1709.07871)

---

> > ### Comment · AnonReviewer2 · 2018-11-27
> > **revised my score.**
> >
> > Thanks for your responses.
> > Yes, I was referring to use the gated attention approach to use as another baseline and use different metric (rather than data efficiency ) to evaluate this framework.
> >
> > Most of my concerns were addressed. As a result, I've updated my score. That said, I am hoping that the authors will add more tasks and metrics to evaluate this platform. Data efficiency metric is good but not enough.

---

### Official Review · AnonReviewer1 · 2018-11-03
**Studies grounded language learning with a human in the loop by removing the human (and natural language)**

**Rating:** 6
**Confidence:** 4

**Review:**

This paper focuses on grounded language learning with a human in the loop, in the sense where the language is synthetic, the environment is a 2D grid world, and the human is a simulated human teacher implemented using heuristics. This setup is dubbed the BabyAI platform, and includes curriculum learning over 19 levels of increasing difficulty.

Overall, the BabyAI platform is conceptually similar to numerous previous works that seek to learn grounded language in simulation environments. These efforts differ along various axes, for example visual realism, 2D vs 3D, partially vs. fully observed, different tasks, world-state manipulation or not, etc. The main original aspect of the BabyAI platform is the simulated human-teacher.

Strengths
- Learning with a human in the loop is an extremely important problem to study, although currently efforts are hampered by cost, lack or reproducibility, and the sample inefficiency of existing learning methods. This paper addresses all three of these issues, albeit by removing the human and natural language. This is simultaneously the greatest weakness of this approach. The contribution of this paper therefore rests on the quality/interestingness/utility of the provided synthetic language and the synthetic teacher.
- Fortunately, the synthetic language does exhibit interesting compositional properties, it is readily extensible, it has the appealing property that it can be readily interpreted as a subset of english, and it is accompanied by a verifier to check if the specified actions were completed.

Weaknesses
- If the ultimate goal is learning with a human in the loop, the usefulness of the synthetic teacher is not clear, particularly as it is apparently easier to imitate from an RL trained agent than the teacher. The explanation 'This can explained by the fact that the RL expert has the same neural network architecture as the learner' does no seem obvious to me.
- Regarding the human in the loop, since this is aspirational and not an aspect of the paper, the title of the paper does not seem reflective of its content (even with the 'First steps' qualifier).
- If the main unique aspect is the simulated human-teacher, it is not clear why it is necessary to create a new environment, rather than re-using an existing environment. The effect of this is to limit comparisons with recent work and an increasing fragmentation of research across tasks that are related but can’t be compared.

Summary:
This paper represents an important direction, in that it provides a testbed for studying the sample efficiency of grounded language learning in a simplified (yet still challenging and compositional) environment. I believe the environment and the provided synthetic language and verifier will prove useful to the community, and despite some reservations about the title and the simulated human-teacher, I recommend acceptance.

---

> ### Author Response · Authors · 2018-11-15
> **Response to Reviewer 1 (part 1 of 2)**
>
> We thank Reviewer 1 (R1) for their careful and detailed review of the paper. In our response we will try to justify the choice of the title, explain why we believe that building a new environment was warranted, and also discuss the difference between imitation learning results obtained with the heuristic expert and an RL-trained agent.
>
> Reviewer 1 has suggested that our aspiration to make progress towards human in the loop training may be insufficient to use the phrase “human in the loop” in the title. With all due respect we would like to argue that the title of a research paper should inform the reader of the high-level goals that are being pursued in the presented research effort. Since the goal of BabyAI platform is to support tangible steps towards human in the loop training, we think that the title “First Steps Towards Grounded Language Learning With a Human In the Loop” is sufficiently accurate. We are open to a continuation of this discussion, but so far our understanding is having “First Steps Towards …” should make it sufficiently clear that training with a human in the loop is something that we aspire to, and not necessarily something that we can already demonstrate.
>
> We thank Reviewer 1 for pointing out that a further discussion of why the synthetic teacher is useful even though an RL-trained agent is easier to imitate may be necessary. The main reason why we chose to build the heuristic expert is that, although RL training did well on some of the simpler levels, it struggled to reach a high success rate on the harder levels. A secondary reason is that, in order to allow further investigations to DAGGER and other more advanced interactive teaching methods, we wanted to have a teacher which could give advice to a learner on which action to take from any state. Unfortunately, RL agents struggle to do this in practice. They generalize poorly to states which they do not normally visit.
>
> As for why RL agents are easier to imitate than the heuristic expert, this is likely because the policy implemented by the RL expert is easier for a neural network to implement. RL is an optimization technique. By design, it attempts to adjust the weights of a neural network so as to maximize the reward obtained on a given problem. In other words, RL will try to find a policy which is the best (in terms of both performance and learnability) for the expert’s neural network. Thus, it may be more natural for a learner that has the same neural network as the  expert to imitate such a policy rather than imitating a computer program, such as our heuristic expert. Informally, we found that the RL-trained policy is more reactive (i.e. based on the current/recent observations), whereas the heuristic expert takes advantage of its perfect memory. In the view of the fact that training RL agents for harder levels is extremely hard, we believe that having a heuristic expert that can solve all levels is highly useful.
>
> (see part 2 for continuation)

---

> > ### Author Response · Authors · 2018-11-15
> > **Response to Reviewer 1 (part 2 of 2)**
> >
> > (see part 1 for the beginning)
> >
> > We fully agree with Reviewer 1 that there are a number of existing options in the space of environments for instruction following. We have examined these options before embarking on this project, and determined that none of them provided the specific combination of features that we wanted, along with a systematically designed language. To the best of our knowledge, the environment we have created is unique in a number of important ways.
> >
> > We chose a gridworld rather than a 3D environment such as in [1, 2] because we wanted an experimental setup that was fast, lightweight, and easy to modify. Using a gridworld means we can run simulations at several thousands of frames per second on a single computer, and train with larger batch sizes. Even so, on some of the more difficult BabyAI levels, training time can take up to a week on a modern GPU. Had we used a 3D environment which was more computationally expensive, the training time requirements would have put this line of research out of our reach.
> >
> > There are already existing options in terms of gridworld packages, such as MazeBase [3] and PyCoLab [4]. However, we wanted a environments that are partially observable, and feature language. Had we used these packages, we would have had to extensively modify them, thereby making any results incomparable to the existing literature. The closest thing to our setup, that we are aware of, is the Crafting 2D env used in the Policy Sketches paper [5]. This environment is interesting, but a quick inspection of the repository will reveal that it is a bare source code dump, with no documentation whatsoever, no installation script, and no maintenance commits in the last two years. This environment is also not compatible with OpenAI Gym.
> >
> > In designing our environment, we wanted a principled approach towards language design, rather than something ad-hoc based on patterns (hence the BNF grammar). We also attempted a principled segmentation of levels in terms of competencies required to solve them. We also believe that ability to scale up/down the difficulty of levels by adjusting various parameters in a fine-grained manner is important to enable curriculum learning, and for research in general, because it can help us establish precisely which aspects of the environment make learning more difficult. Our environment was designed with this in mind.
> >
> > [1] Grounded Language Learning in a Simulated 3D World (https://arxiv.org/abs/1706.06551)
> > [2] Project Malmo (https://github.com/Microsoft/malmo)
> > [3] MazeBase (https://github.com/facebook/MazeBase)
> > [4] PyCoLab (https://github.com/deepmind/pycolab)
> > [5] Policy Sketches implementation (https://github.com/jacobandreas/psketch)

---

### Public Comment · (anonymous) · 2018-11-07
**Format of the observation**

Very interesting work!

In Appendix A.4, you describe how the observations are encoded: 'Each tile is encoded using 3 integer values: one describing
the type of object contained in the cell, one describing its color, and a flag indicating whether doors
are open or closed.'. I was wondering whether these are given to the agent as actual integers, or they are one-hot encoded first. Could you please comment on this?

---

> ### Author Response · Authors · 2018-11-12
> **Clarification on Observation Format**
>
> Thank you for your question. The agent indeed receives 3 integers for each tile. In our preliminary investigations we tried converting these to one-hot embeddings first, but we did not observe a big difference in the results.
>
> Please let us know if you have any further questions.

---

### Author Response · Authors · 2018-11-23
**Paper updated based on reviewer feedback**

Dear reviewers, thank you for your useful feedback. Following your recommendations, we have made the following changes to improve our paper:

1. We now use the term “bot” as often as possible, instead of heuristic expert.

2. We have highlighted the connection between gated attention and FiLM when describing our model in Section 4.1.

3. A paragraph that discusses other existing simulation package was added to the related work, with a citation of the AI2-Thor and Matterport paper. We have also added a better explanation of why we have built MiniGrid.

4. In the experiments section, we highlight the counter-intuitive finding Table 4, where data requirements for GoToObjMaze "With Pretraining" are greater than "Without Pretraining".

5. A detailed explanation of how instructions are translated into an internal system of subgoals by the bot was added to Appendix B.

We hope that these changes effectively address your concerns,

Kindest regards,

- The BabyAI team

---

### Author Response · Authors · 2018-12-13
**Paper Title Change**

Hello,

Following internal discussions, we have decided, in agreement with two reviewers, to change the title of the paper. The new title will be: "BabyAI: A Platform to Study the Sample Efficiency of Grounded Language Learning".

Kindest regards,

- The BabyAI team

---

### Meta-Review · Area_Chair1 · 2018-12-14
**a new platform that supports interactive and grounded language learning**

**Confidence:** 3
**Recommendation:** Accept (Poster)

**Metareview:**

This paper presents "BabyAI", a research platform to support grounded language learning. The platform supports a suite of 19 levels, based on *synthetic* natural language of increasing difficulties. The platform uniquely supports simulated "human-in-the-loop" learning, where a human teacher is simulated as a heuristic expert agent speaking in synthetic language.

Pros:
A new platform to support grounded natural language learning with 19 levels of increasing difficulties. The platform also supports a heuristic expert agent to simulate a human teacher, which aims to mimic "human-in-the-loop" learning. The platform seems to be the result of a substantial amount of engineering, thus nontrivial to develop. While not representing the real communication or true natural language, the platform is likely to be useful for DL/RL researchers to perform prototype research on interactive and grounded language learning.

Cons:
Everything in the presented platform is based on synthetic natural language. While the use of synthetic language is not entirely satisfactory, such limit is relatively common among the simulation environments available today, and lifting that limitation is not straightforward. The primary contribution of the paper is a new platform (resource). There are no insights or methods.

Verdict:
Potential weak accept. The potential impact of this work is that the platform will likely be useful for DL/RL research on interactive and grounded language learning.